# Detecting and describing stability and change in COVID-19 vaccine receptibility in the United Kingdom and Ireland

Philip Hyland[1], Frédérique Vallières[2], Todd K. Hartman[3], Ryan McKay[4], Sarah Butter[5]*, Richard P. Bentall[5], Orla McBride[6], Mark Shevlin[6], Kate Bennett[7], Liam Mason[8], Jilly Gibson-Miller[5], Liat Levita[5], Anton P. Martinez[5], Thomas V. A. Stocks[5], Thanos Karatzias[9], Jamie Murphy[6]

1 Department of Psychology, Maynooth University, Maynooth, Ireland, 2 Trinity Centre for Global Health, Trinity College Dublin, Dublin, Ireland, 3 Sheffield Methods Institute, University of Sheffield, Sheffield, England, 4 Department of Psychology, Royal Holloway, University of London, London, England, 5 Department of Psychology, University of Sheffield, Sheffield, England, 6 School of Psychology, Ulster University, Ulster, Northern Ireland, 7 School of Psychology, University of Liverpool, Liverpool, England, 8 Clinical, Education & Health Psychology, University College London, London, England, 9 School of Health and Social Care, Edinburgh Napier University, Edinburgh, Scotland

* s.butter@sheffield.ac.uk

**Data Availability Statement:** All data files relating to this manuscript are available from the C19PRC Study Open Science Framework page (https://osf.io/ugwdz/files/).

## Abstract

COVID-19 continues to pose a threat to global public health. Multiple safe and effective vaccines against COVID-19 are available with one-third of the global population now vaccinated. Achieving a sufficient level of vaccine coverage to suppress COVID-19 requires, in part, sufficient acceptance among the public. However, relatively high rates of hesitance and resistance to COVID-19 vaccination persists, threating public health efforts to achieve vaccine-induced population protection. In this study, we examined longitudinal changes in COVID-19 vaccine acceptance, hesitance, and resistance in two nations (the United Kingdom and the Republic of Ireland) during the first nine months of the pandemic, and identified individual and psychological factors associated with consistent non-acceptance of COVID-19 vaccination. Using nationally representative, longitudinal data from the United Kingdom (UK; N = 2025) and Ireland (N = 1041), we found that (1) COVID-19 vaccine acceptance declined in the UK and remained unchanged in Ireland following the emergence of approved vaccines; (2) multiple subgroups existed reflecting people who were consistently willing to be vaccinated ('Accepters': 68% in the UK and 61% in Ireland), consistently unwilling to be vaccinated ('Deniers': 12% in the UK and 16% in Ireland), and who fluctuated over time ('Moveable Middle': 20% in the UK and 23% in Ireland); and (3) the 'deniers' and 'moveable middle' were distinguishable from the 'accepters' on a range of individual (e.g., younger, low income, living alone) and psychological (e.g., distrust of scientists and doctors, conspiracy mindedness) factors. The use of two high-income, Western European nations limits the generalizability of these findings. Nevertheless, understanding how receptibility to COVID-19 vaccination changes as the pandemic unfolds, and the factors that distinguish and characterise those that are hesitant and resistant to vaccination is helpful for public health efforts to achieve vaccine-induced population protection against COVID-19.

**Funding:** UK Research and Innovation/Economic and Social Research Council funding for the UK strand of this study was obtained in May 2020 (grant number ES/V004379/1) and awarded to RPB, TKH, LL, JGM, MS, JM, OM, KB and LM. The Irish strand of this study was funded by the Health Research Board and the Irish Research Council under the COVID-19 Pandemic Rapid Response Funding Call [COV19-2020-025] awarded to PH. The funders had no role in study design, data collection and analysis, decision to publish, or preparation of the manuscript.

**Competing interests:** The authors declare the following financial interests/personal relationships which may be considered as potential competing interests: Richard Bentall reports financial support was provided by UK Research and Innovation. Philip Hyland reports financial support was provided by Health Research Board. This does not alter our adherence to PLOS ONE policies on sharing data and materials.

## Introduction

The rapid development of safe and effective vaccines against Coronavirus Disease (COVID-19) represents one of the greatest collaborative scientific achievements of our lifetime. As of August 2021, four vaccines have been authorised by the European Medicines Agency, three have been authorized for emergency use by the United States Food and Drug Administration, and 99 are undergoing clinical trials on humans [1]. Just under five billion vaccines doses have been administered, globally, meaning that 31% of the world's population have been vaccinated and it is estimated that 75% of the world's population will be vaccinated by February 2021 [2]. Sufficient uptake of COVID-19 vaccines not only requires the coordinated action of governments, communities, and individuals alike to ensure adequate vaccine *delivery* (e.g., via production, logistics, procurement, financing, and service delivery components of the health system), but also to ensure vaccine *receptibility*.

COVID-19 vaccine acceptance rates across the world range from lows of 24% in Kuwait and 44% in Lebanon to highs of 88% in China and 91% in India [3–5]. Concurrently, rising rates of vaccine hesitancy, whereby an individual delays or refuses vaccination despite the availability of inoculation services [6], remains one of the greatest global health threats listed by the World Health Organization [7]. As the term implies, however, vaccine hesitancy is not immutable, and individual attitudes towards a specific vaccine can change over time as a function of a wide-range of interdependent individual, social, and vaccination-specific factors [8, 9] including, but not limited to, perceptions of susceptibility to pathogen exposure [10], severity of illness [11], perceived vaccine safety and efficacy [11, 12], and recency of vaccine development [13]. Accordingly, some have suggested that vaccine hesitancy is better conceptualised as existing on a continuum and bookended by 'decliners' and 'accepters', or those who completely reject or accept all vaccines, respectively [14]. Levels of COVID-19 vaccine acceptability have fluctuated considerably throughout the pandemic. Most recent data from the global survey of knowledge, attitudes, and practices around COVID-19 (KAP COVID-19)—which has reached over 1.7 million people in 67 countries across as many as 19 waves of data collection in some contexts—indicates that only 63% of individuals would accept a COVID-19 vaccine as of the 31st of January 2021 [15]. Encouragingly, however, these same data suggest that willingness to be vaccinated has increased in nations that have successfully launched COVID-19 vaccination programmes (e.g., the United Kingdom [UK]).

Previous work carried out by our group, the COVID-19 Psychological Research Consortium, found that resistance to COVID-19 vaccination in the UK and the Republic of Ireland is associated with distrust of experts and authority figures (i.e., scientists, health care professionals, and government), stronger religious, conspiratorial, and paranoid beliefs, a higher internal locus of control, preference for hierarchically structured and authoritarian societies, anti-migrant views, lower levels of agreeableness, conscientiousness, and emotional stability [16]. Similarly, the 'attitude roots' model of science rejection proposes that conspiratorial beliefs, disgust sensitivity, trait reactance—as a motivational state that arises when people feel that their behavioural freedom has been threatened or taken away [17]—and hierarchical worldviews are central to understanding individual differences in vaccine resistant attitudes [18–21]. Thus, understanding the individual factors, including psychological dispositions, that predict whether vaccine hesitant individuals change their minds about COVID-19 vaccination, as well as the factors that might predict a move towards acceptance or resistance over time is paramount, albeit currently less well understood [9].

In light of these existing gaps, the current study was planned with three primary objectives. The first was to examine changes in COVID-19 vaccine acceptance, hesitance, and resistance in the Irish and UK adult populations across four time periods (Waves) during the first nine

months of the global pandemic. We have previously reported on the changes in these populations across the first three waves of the survey (i.e., March-April, April-May, and July-August 2020) [22]; however, as these data were obtained prior to the development of safe and effective vaccines for COVID-19, our focus in this study is on *changes* from Wave 3 (July/August 2020) to Wave 4 (November/December 2020) when populations transitioned from having to contemplate a hypothetical vaccine to considering an actual, available vaccine.

Understanding that people's willingness to accept a COVID-19 vaccine may fluctuate over time, our second objective was to determine if there were multiple groups in each sample with distinct probabilities of accepting a COVID-19 vaccine over time. We hypothesised that there would be two stable groups in each sample: one representing people with consistently high probabilities of accepting a COVID-19 vaccine ('Accepters'), and the other representing people with consistently low probabilities of accepting a COVID-19 vaccine ('Deniers'). Additionally, we expected to identify a group (or groups) in each sample with fluctuating probabilities of accepting a COVID-19 vaccine; a group that have often been termed the movable middle.

Finally, we sought to identity key sociodemographic and psychological factors that were associated with belonging to any group that was not consistent in their acceptance of a COVID-19 vaccine. Our intention with the second and third objectives was to develop a comprehensive understanding of the people who were *not consistent* in their willingness to accept a COVID-19 vaccine so that targeted and effective public health strategies could be developed to reach those who can still change their minds.

## Material and methods

### Participants and procedures

This study is based on data from the Irish and UK strands of the COVID-19 Psychological Research Consortium (C19PRC) study. The C19PRC study was established to track the social, political, economic, and mental health effects of the COVID-19 pandemic on society. Data for this study were collected at four assessment points during the first nine months of the COVID-19 pandemic. Wave 1 data were collected in the UK between March 23rd and 28th, 2020, and in Ireland between March 30th and April 5th, 2020. These dates coincided with the initial public health lockdown measures in the respective countries. Wave 2 data were collected in the UK from April 22nd to May 1st, 2020, and in Ireland from April 30th to May 19th, 2020. Wave 3 data were collected in the UK from July 9th to July 23rd, 2020, and in Ireland from July 16th to August 8th, 2020. Finally, Wave 4 data were collected in the UK from November 25th to December 22nd, 2020, and in Ireland from December 2nd to December 22nd, 2020.

The UK and Irish samples were collected using a non-probability Internet panel survey design. The survey research company Qualtrics was employed to recruit participants from traditional, actively managed, double-opt-in research panels via email, SMS, or in-app notifications. Inclusion criteria for both samples were that respondents were aged 18 years or older, residing in the UK or Ireland, respectively, and capable of completing the survey in English. Ethical approval was granted by the research ethics committees at the University of Sheffield (Reference number: 033759), Ulster University (Reference number: 230320), and Maynooth University (Reference number: SRESC-2020-2402202). Participants were remunerated by Qualtrics, and informed electronic consent was obtained from all participants. Quota sampling methods were used at Wave 1 to generate samples that represented the general adult populations of both nations. In the UK, the sample was recruited to match known population quotas for sex, age, and income distributions. In Ireland, the sample was recruited to match known population quotas for sex, age, and regional distribution. Further details regarding the UK and Irish samples, including evidence of their representativeness, are presented elsewhere [23–25].

As described in an earlier study [16], power analyses to determine optimal sample sizes were calculated to detect common mental health disorders such as Major Depressive Disorder and Posttraumatic Stress Disorder. Sample size calculations were performed to detect a disorder with a 4% prevalence rate, with a precision of 1%, and 95% confidence levels. This resulted in a required sample size of 1,476. As Qualtrics could only guarantee a sample size of 1,000 participants in Ireland, this was set as the target sample size in Ireland. Holding all other parameters in the sample size calculation equal, this sample size resulted in a precision of 1.21%. Given the substantially larger population of the UK and thus the availability of a larger pool of potential participants, we set a target sample size of 2,000 people.

At Wave 1, the sample size in the UK was 2,025 and 1,041 in Ireland. The sociodemographic characteristics for both samples at Wave 1 are reported in Table 1. In the UK, the recontact rate was 69% ($n = 1406$) at Wave 2, 58% ($n = 1166$) at Wave 3, and 63% ($n = 1271$) at Wave 4. Those who responded at each wave significantly differed ($p < .05$) from non-responders on a range of sociodemographic variables including being older, male, living with fewer adults, higher income earners, born in the UK, not living in a city, having a post-secondary education, and not having a suspected or confirmed COVID-19 infection.

In Ireland, the recontact rate was 49% ($n = 506$) at Wave 2, 51% (n = 534) at Wave 3, and 40% (n = 416) at Wave 4. Respondents significantly differed ($p < .05$) from non-responders by being older, more likely to have been born in Ireland, not living in a city, to have a pre-existing health condition, and not having a suspected or confirmed COVID-19 infection. Management of missing data is outlined in the data analysis section.

## Materials

**COVID-19 vaccination status.** In the UK and Irish samples, participants were asked the following question at Waves 1, 2, and 3: 'If a new vaccine were to be developed that could prevent COVID-19, would you accept it for yourself?' At Wave 4, participants in both samples were asked: 'Multiple vaccines for COVID-19 have now been developed. Will you take a vaccine for COVID-19 when it becomes available to you?' The response options at all times were 'Yes', 'Maybe', and 'No'. Those who answered 'Yes' were classified as 'vaccine accepting', those who responded 'Maybe' were classified as 'vaccine hesitant', and those who responded 'No' were classified as 'vaccine resistant'.

**Sociodemographic, political, and health indicators (Measured at Wave 1).** The sociodemographic, political, and health indicator variables used in this study were identical to those utilized in our previous study [16], and all are listed in Table 1. For analytical purposes, several of these variables were recoded. Living location was recoded to represent city dwelling vs. non-city dwelling; education status was recoded to represent post-secondary education vs. non-post-secondary education; employment status was recoded to represent unemployed vs. all other options; and religion was recoded to represent any religious identification vs. atheist or agnostic. Additionally, due to limited numbers in various subgroups, ethnicity was recoded to represent self-identified Irish ethnicity vs. non-Irish ethnicity in the Irish sample.

**Psychological indicators (Measured at Wave 1).** *Personality traits.* The Big-Five Inventory (BFI-10) [26] measures the traits of openness to experience, conscientiousness, extraversion, agreeableness, and neuroticism. Each trait is measured by two items using a five-point Likert scale that ranges from 'strongly disagree' (1) to 'strongly agree' (5). Higher scores reflect higher levels of each personality trait, and Rammstedt and John [26] reported good reliability and validity for the BFI-10 scale scores. Internal reliability coefficients are not provided as this scale measures each trait using only two items, and it is well documented that coefficient alpha is inappropriate and meaningless for two-item scales [27].

**Table 1. Sociodemographic characteristics of the Irish and UK samples.**

| Ireland (N = 1041) | % | UK (N = 2025) | % |
|---|---|---|---|
| **Sex** | | **Sex** | |
| Female | 51.5 | Female | 51.7 |
| Male | 48.2 | Male | 48.3 |
| **Age** | | **Age** | |
| 18–24 | 11.1 | 18–24 | 12.1 |
| 25–34 | 19.2 | 25–34 | 18.8 |
| 35–44 | 20.6 | 35–44 | 17.4 |
| 45–54 | 15.9 | 45–54 | 20.2 |
| 55–64 | 21.0 | 55–64 | 17.2 |
| 65+ | 12.2 | 65+ | 14.2 |
| **Born in Ireland** | 70.7 | **Born in UK** | 90.6 |
| **Region of Ireland** | | **Region of UK** | |
| Leinster | 55.3 | England | 86.9 |
| Munster | 27.3 | Scotland | 7.8 |
| Connacht | 12.0 | Wales | 3.1 |
| Ulster | 5.4 | Northern Ireland | 2.3 |
| **Ethnicity** | | **Ethnicity** | |
| Irish | 74.8 | White British/Irish | 85.5 |
| Irish Traveller | 0.3 | White non-British/Irish | 5.7 |
| Other White background | 17.3 | Indian | 2.0 |
| African | 1.9 | Pakistani | 1.3 |
| Other Black background | 0.3 | Chinese | 0.9 |
| Chinese | 0.4 | Afro-Caribbean | 0.6 |
| Other Asian | 3.2 | African | 1.3 |
| Mixed Background | 1.8 | Arab | 0.1 |
| | | Bangladeshi | 0.3 |
| | | Other Asian | 0.5 |
| **Living location** | | **Living location** | |
| City | 24.5 | City | 24.6 |
| Suburb | 18.1 | Suburb | 28.2 |
| Town | 26.8 | Town | 30.6 |
| Rural | 28.8 | Rural | 16.5 |
| **Highest Education** | | **Highest Education** | |
| No qualification | 1.2 | No qualifications | 2.9 |
| Finished mandatory schooling | 6.4 | O-level/GCSE or similar | 19.0 |
| Finished secondary school | 22.4 | A-level or similar | 18.1 |
| Undergraduate degree | 22.5 | Diploma | 5.6 |
| Postgraduate degree | 19.8 | Undergraduate degree | 28.2 |
| Other technical qualification | 27.9 | Postgraduate degree | 15.6 |
| | | Technical qualification | 9.3 |
| | | Other | 1.3 |
| **2019 income** | | **2019 income** | |
| 0-€19,999 | 24.6 | £0-£15490 | 20.2 |
| €20,000-29,999 | 21.3 | £15,491-£25,340 | 20.2 |
| €30,000-€39,999 | 19.5 | £25,341-£38,740 | 19.0 |
| €40,000-€49,999 | 12.7 | £38,741-£57,930 | 20.2 |
| €50,000+ | 21.9 | £57,931+ | 20.2 |

(*Continued*)

**Table 1.** (Continued)

| Ireland (N = 1041) | % | UK (N = 2025) | % |
|---|---|---|---|
| **Employment status** | | **Employment status** | |
| Full-time (self/)employed | 43.3 | Full-time (self/)employed | 48.8 |
| Part-time (self/)employed | 15.7 | Part-time (self/)employed | 15.0 |
| Retired | 15.0 | Retired | 16.5 |
| Unemployed | 8.4 | Unemployed | 11.7 |
| Student | 6.3 | Student | 4.7 |
| Unemployed (disability or illness) | 5.6 | Unemployed (disability or illness) | 3.4 |
| Unemployed due to COVID-19 | 5.7 | | |
| **Religious identification** | | **Religious identification** | |
| Christian | 69.8 | Christian | 50.4 |
| Muslim | 1.6 | Muslim | 3.0 |
| Jewish | 0.2 | Jewish | 0.8 |
| Hindu | 1.1 | Hindu | 0.6 |
| Buddhist | 0.6 | Buddhist | 0.8 |
| Sikh | 0.1 | Sikh | 0.5 |
| Other religion | 3.8 | Other | 6.0 |
| Atheist | 15.3 | Atheist | 25.4 |
| Agnostic | 7.5 | Agnostic | 12.5 |
| **Lone adult in household** | | **Lone adult in household** | |
| Yes | 18.4 | Yes | 22.4 |
| **Children in the household** | | **Children in the household** | |
| Yes | 39.7 | Yes | 29.2 |
| **Physical health problem** | 16.7 | **Physical health problem** | 15.4 |
| **Pregnant** | 4.0 | **Pregnant** | 3.8 |
| **COVID-19 infection—self** | 2.3 | **COVID-19 infection—self** | 2.4 |
| **COVID-19 infection—other** | 6.7 | **COVID-19 infection—other** | 5.5 |
| **Mental health treatment** | 33.0 | **Mental health treatment** | 32.0 |
| **Voting behaviour** | | **Voting behaviour** | |
| Fine Gael | 17.4 | Conservative Party | 42.0 |
| Fianna Fail | 11.9 | Labour Party | 28.4 |
| Sinn Fein | 22.8 | Liberal Democrats | 10.3 |
| Green Party | 5.4 | Green Party | 5.0 |
| Labour Party | 3.8 | Other nationalist parties | 5.1 |
| Other left-wing parties | 6.1 | Other unionist parties | 3.3 |
| Independent | 8.1 | Other party | 2.8 |
| Did not vote | 24.5 | Did not vote | 4.2 |

*Locus of control.* The Locus of Control Scale (LoC) [28] measures internal (e.g., 'My life is determined by my own actions') and external locus of control. The latter has two components, 'Chance' (e.g., 'To a great extent, my life is controlled by accidental happenings') and 'Powerful Others' (e.g., 'Getting what I want requires pleasing those people above me'). Each subscale was measured using three questions and a seven-point Likert scale that ranges from 'strongly disagree' (1) to 'strongly agree' (7). Higher scores reflect higher levels of each construct. The internal reliabilities of the Internal and Chance subscale scores in the Irish sample were slightly lower than desirable ($\alpha$ = .67 &.63, respectively) but somewhat stronger for the UK sample ($\alpha$ = .71 &.70, respectively), while those for the Powerful Others subscale scores were good in both samples (Ireland: $\alpha$ = .78; UK: $\alpha$ = .85).

*Analytical/reflective reasoning*. The Cognitive Reflection Task (CRT) [29] is a three-item measure of analytical reasoning where respondents are asked to solve logical problems designed to hint at intuitively appealing but incorrect responses. The response format was multiple choice with three foil answers (including the hinted incorrect answer), as recommended by Sirota and Juanchich [30]. The internal reliabilities of the CRT scores in the Irish and UK samples were $\alpha = .67$ and $\alpha = .69$, respectively.

*Altruism*. The Identification with all Humanity scale (IWAH) [31] is a nine-item scale. Respondents are asked to respond to three statements with reference to three groups; people in my community, people from Ireland/ the UK, and all humans everywhere. The three statements were presented to respondents separately for each of the three groups, as follows: (1) How much do you identify with (feel a part of, feel love toward, have concern for) . . .? (2) How much would you say you care (feel upset, want to help) when bad things happen to . . .? And, (3) When they are in need, how much do you want to help. . .? Response scale ranged from 1 'not at all' to 5 'very much'. Higher scores reflect greater identification with others, care for others, and a desire to help others. The internal reliabilities of each subscale of the IWAH in both the Irish and UK samples were excellent (identification with others $\alpha = .79$ &.81; care for others $\alpha = .88$ &.89; desire to help others $\alpha = .86$ &.88, respectively).

*Conspiracy beliefs*. The Conspiracy Mentality Scale (CMS) [32] measures conspiracy mindedness using five items with each scored on an 11-point scale (1 = 'Certainly not 0%' to 11 = 'Certainly 100%'). Items include, 'I think that many very important things happen in the world, which the public is never informed about', and 'I think that there are secret organizations that greatly influence political decisions'. The internal reliability of the CMS in both the Irish and UK samples was good ($\alpha = .84$ &.85, respectively).

*Paranoia*. The five-item persecution subscale from the Persecution and Deservedness Scale was used [33]. Participants rate their agreement with statements such as "I'm often suspicious of other people's intentions towards me" and "You should only trust yourself." Response options ranged from 'strongly disagree' (1) to 'strongly agree' (5) with higher scores reflecting higher levels of paranoia. The psychometric properties of the scale scores have been previously supported [34], and the internal reliability in both the Irish and UK samples was good ($\alpha = .83$ &.86, respectively).

*Trust*. Respondents were asked to indicate the level of trust they have in political parties, Parliament, the government, the police, the legal system, scientists, and doctors and other health professionals. Responses were scored on a five-point Likert scale ranging from 'do not trust at all' (1) to 'completely trust' (5). For this study, responses to the first five institutions were summed to generate a total score for 'trust in the state'. Responses to the final two questions were summed to generate a total score for 'trust in scientists and doctors/health professionals'.

*Authoritarianism*. The Very Short Authoritarianism Scale [35] includes six items assessing agreement with statements such as: 'It's great that many young people today are prepared to defy authority' and 'What our country needs most is discipline, with everyone following our leaders in unity'. All items were scored on a five-point Likert scale ranging from 'strongly disagree' (1) to 'strongly agree' (5), with higher scores reflecting higher levels of authoritarianism. The internal reliability of the scale scores in the Irish sample was lower than desirable ($\alpha = .58$) but somewhat stronger for the UK sample ($a = .65$).

*Social dominance*. Respondents' levels of social dominance orientation were assessed using the eight-item Social Dominance Scale [36]. Respondents were asked the extent to which they opposed/favoured statements such as: 'An ideal society requires some groups to be on top and others to be on the bottom'; 'Some groups of people are simply inferior to other groups'; and 'We should do what we can to equalize conditions for different groups'. Responses were scored

using a 5-point Likert scale ranging from 1 'Strongly oppose' to 5 'Strongly Favour'. Ho and colleagues demonstrated that the scale had good criterion and construct validity [36]. The internal reliability of the scale scores in both the Irish and UK samples was good ($\alpha$ = .79 &.82, respectively).

*Attitude towards migrants*. Two items assessing respondents' attitudes towards migrants were taken from the British Social Attitudes Survey 2015 [37]. These were, (1) 'Would you say it is generally bad or good for the UK's economy that migrants come to the UK from other countries?' (scored on a 10-point scale ranging from 1 'extremely bad' to 10 'extremely good'), and (2) 'Would you say that the UK's cultural life is generally undermined or enriched by migrants coming to live here from other countries?' (scored on a 10-point scale ranging from 1 'undermined' to 10 'enriched'). These items were phrased appropriately for use with the Irish sample.

## Data analysis

The first objective was assessed by means of structural equation modelling (SEM). A SEM approach was used so that missing data could be most effectively managed using full information robust maximum likelihood estimation (MLR) [38]. This approach is helpful because it means that all available information at Wave 1 is used to estimate missingness at future waves, thus ensuring minimal loss of statistical power or sample representativeness. This method of estimation can also handle non-normally distributed variables [39]. This analytic process involved three steps. First, a 'null' model was specified where the proportions (e.g., in vaccine acceptance, hesitance, and resistance—all are estimated individually) at Waves 1–4 were constrained to be equal. Second, an 'alternative' model was specified where the proportions were freely estimated at each wave. These models differed by three degrees of freedom and significant differences in model fit were tested using a loglikelihood ratio test (LRT), which follows a chi-square ($\chi^2$) distribution. Third, post-hoc pairwise comparisons were tested using a Wald $\chi^2$ test.

The second objective was assessed using latent class analysis (LCA). Responses to the question about willingness to accept a COVID-19 vaccine (0 = Yes, 1 = Maybe, 2 = No) at Waves 1–4 were used as the observed indicators in the model. To understand the probability of consistent acceptance of a COVID-19 vaccine across time, we focused our interpretations on the probability of the first response (i.e., 'Yes') within each class. Models with one to six classes were estimated in the Irish and UK samples using MLR. To avoid solutions based on local maxima, 500 random starting values and 50 final stage optimizations were used. The relative fit of these models was compared using three information theory based fit statistics: the Akaike Information Criterion (AIC) [40], the Bayesian Information Criterion (BIC) [41] and the sample size adjusted Bayesian Information Criterion (ssaBIC) [42]. The solution with the lowest value of these statistics is deemed superior, or if no minimum is found then the 'diminishing gains in model fit' for additional classes can be examined [43]. Simulation studies suggest that the BIC is optimal for identifying the correct number of classes [44]. Additionally, the Lo-Mendell-Rubin adjusted likelihood ratio test (LMR-A) [45] was used to compare models with increasing numbers of latent classes. When a non-significant value occurs, the model with one fewer class should be accepted. Model convergence, replication of the log-likelihood, entropy values, the plausibility of the model estimates, and the interpretability of the model solutions were also used to determine the optimal solution.

The third objective was assessed by adding the demographic and psychological predictor variables to the best fitting LCA models in the Irish and UK samples, respectively. A 3-step approach was used so that the inclusion of the predictor variables did not influence the formation of the classes [46].

## Results

### Objective 1: Vaccine acceptance, hesitance, and resistance

From March/April 2020 (Wave 1) to December 2020 (Wave 4) in Ireland, there was evidence of significant change in rates of vaccine acceptance ($\chi^2$ (3, 1030) = 40.12, $p < .001$) and resistance ($\chi^2$ (3, 1030) = 45.34, $p < .001$), but not vaccine hesitance ($\chi^2$ (3, 1030) = 4.41, $p = .220$). From March 2020 (Wave 1) to November/December 2020 in the UK, there was evidence of significant change in rates of vaccine acceptance ($\chi^2$ (3, 2020) = 26.82, $p < .001$), hesitance ($\chi^2$ (3, 2020) = 39.96, $p < .001$), and resistance ($\chi^2$ (3, 2020) = 110.78, $p < .001$). The nature of these changes in both samples are presented in Fig 1, and the pairwise comparisons are presented in Table 2.

In the Irish sample, there were no significant changes in vaccine acceptance ($\chi^2$ (1, 1030) = 0.07, $p = .793$), hesitance ($\chi^2$ (1, 1030) = 0.12, $p = .726$), or resistance ($\chi^2$ (1, 1030) = 0.29, $p = .589$) between Wave 3 and Wave 4. In the UK sample, there was a significant decrease in vaccine acceptance ($\chi^2$ (1, 2020) = 20.06, $p < .001$), no significant change in vaccine hesitance

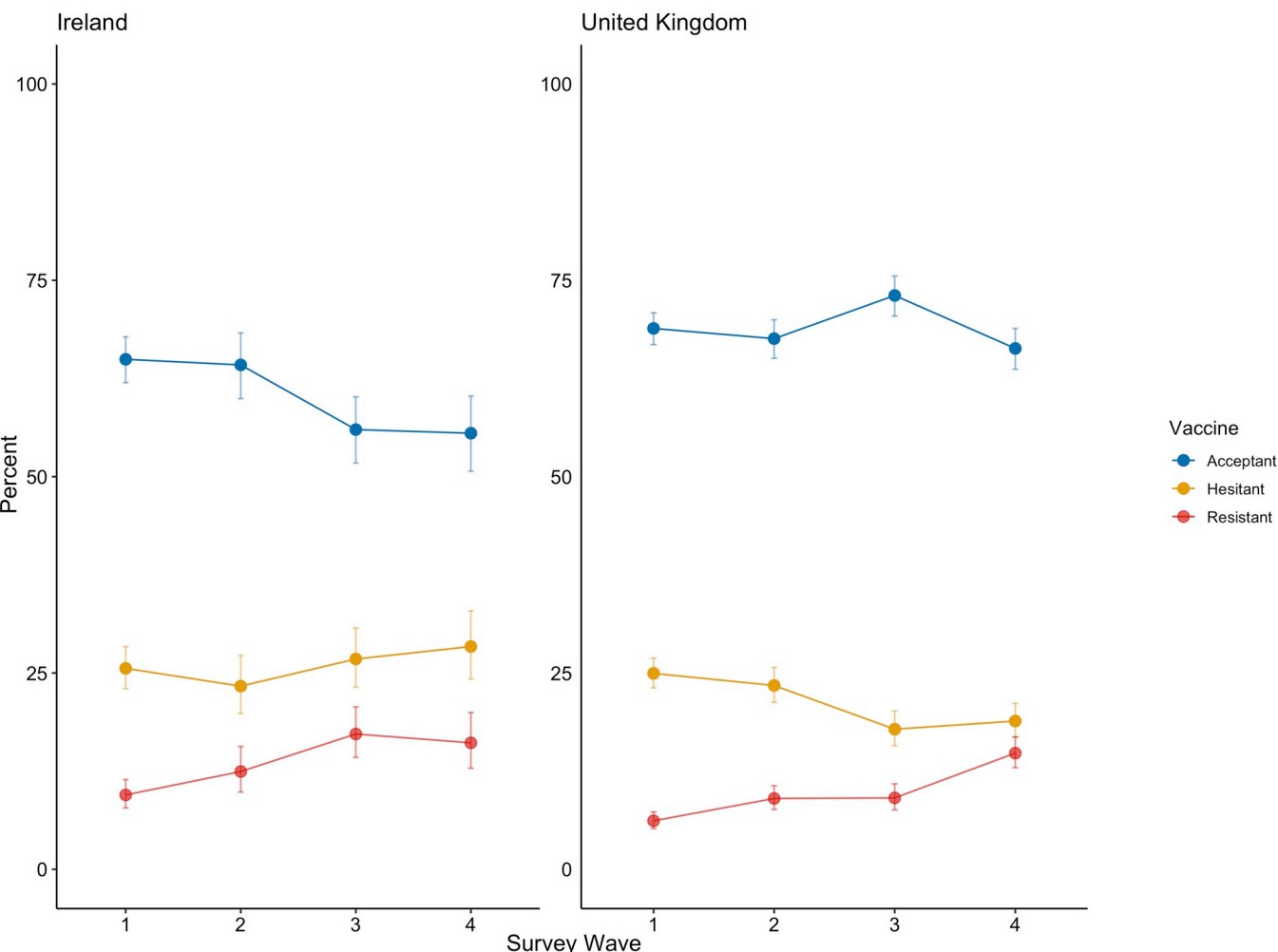

**Fig 1. COVID-19 vaccine acceptance, hesitance, and resistance in the Irish and UK samples.** Data are presented as the proportion of the Irish (n = 1030) and United Kingdom (n = 2000) samples indicating COVID-19 acceptance (blue line), hesitance (orange line), and resistance (red line) across four waves of data collection (Wave 1, March-April 2020, Wave 2 is April-May 2020, Wave 3 is July-August 2020, and Wave 4 is November-December 2020).

**Table 2. Pairwise comparisons for the Irish (N = 1,030) and UK (N = 2,020) samples.**

| | Ireland | | UK | |
|---|---|---|---|---|
| | Wald $\chi^2$ | $p$ | Wald $\chi^2$ | $p$ |
| **COVID-19 Vaccine Acceptance** | | | | |
| Wave 1 vs. Wave 2 | 0.07 | .786 | 3.40 | .065 |
| Wave 1 vs. Wave 3 | 25.23 | <.001 | 3.14 | .077 |
| Wave 1 vs. Wave 4 | 15.82 | <.001 | 5.67 | .017 |
| Wave 2 vs. Wave 3 | 22.74 | <.001 | 14.66 | <.001 |
| Wave 2 vs. Wave 4 | 12.40 | <.001 | 0.59 | .444 |
| Wave 3 vs. Wave 4 | 0.07 | .793 | 20.06 | <.001 |
| **COVID-19 Vaccine Hesitance** | | | | |
| Wave 1 vs. Wave 2 | 1.99 | .158 | 0.95 | .330 |
| Wave 1 vs. Wave 3 | 0.28 | .595 | 29.52 | <.001 |
| Wave 1 vs. Wave 4 | 0.68 | .411 | 21.41 | <.001 |
| Wave 2 vs. Wave 3 | 3.24 | .072 | 22.65 | <.001 |
| Wave 2 vs. Wave 4 | 3.30 | .069 | 13.50 | <.001 |
| Wave 3 vs. Wave 4 | 0.12 | .726 | 0.41 | .520 |
| **COVID-19 Vaccine Resistance** | | | | |
| Wave 1 vs. Wave 2 | 6.04 | .014 | 18.91 | <.001 |
| Wave 1 vs. Wave 3 | 30.63 | <.001 | 26.53 | <.001 |
| Wave 1 vs. Wave 4 | 20.01 | <.001 | 92.23 | <.001 |
| Wave 2 vs. Wave 3 | 11.50 | <.001 | 1.51 | .220 |
| Wave 2 vs. Wave 4 | 5.61 | .018 | 35.32 | <.001 |
| Wave 3 vs. Wave 4 | 0.29 | .589 | 25.28 | <.001 |

Note: $\chi^2$ = chi-square; all Wald $\chi^2$ tests have one degree of freedom.

($\chi^2$ (1, 2020) = 0.41, $p$ = .520), and a significant increase in vaccine resistance ($\chi^2$ (1, 2020) = 25.28, $p$ < .001) between Wave 3 and Wave 4.

## Objective 2: Changing probabilities of vaccine acceptance over time

The full set of latent class analysis (LCA) results for the Irish and UK samples are presented in Table 3. In both samples, iterative models with one to four classes terminated normally, and the loglikelihood values were replicated. Models with more than four classes failed to converge or terminate normally in both samples suggesting that models with more than four classes were not viable representations of the sample data. Overall, the results were similar in the two samples in that the Bayesian Information Criteria (BIC) and sample size adjusted BIC (ssaBIC) values were lowest for the three-class models. The Lo-Mendell-Rubin adjusted likelihood-ratio test (LMR-A) values become non-significant at five classes, which suggests that a four-class model may be optimal; however, the $p$-values for the four-class model were also elevated (Ireland: $p$ = .022; UK: $p$ = .027), suggesting a better fit for the three-class model. Comparing the profiles of the three- and four-class models, a relatively large group of people with high probabilities of accepting a COVID-19 vaccine in the three-class model was differentiated in the four-class model to represent groups with high and moderate-to-high probabilities of vaccine acceptance. Thus, the addition of another class in the four-class model was not qualitatively different from the classes identified in the more parsimonious three-class model. Consequently, based on parsimony, model interpretability, and recognition that BIC is an optimal

**Table 3. Fit indices for latent class models in the Irish and UK samples.**

|  | Log likelihood | AIC | BIC | ssaBIC | LMR-A (p) | Entropy |
|---|---|---|---|---|---|---|
| **Ireland** |  |  |  |  |  |  |
| 1 | -2250.26 | 4516 | 4556 | 4530 | - - | - - |
| 2 | -1959.16 | 3952 | 4036 | 3982 | 573.02 (< .001) | .67 |
| 3 | -1889.46 | 3830 | 3959 | 3876 | 137.22 (< .001) | .67 |
| 4 | -1880.67 | 3831 | 4004 | 3892 | 17.30 (.022) | .64 |
| 5 | -1873.07* | 3834 | 4051 | 3911 | 15.08 (.883) | .56 |
| 6 | -1867.68 | 3841 | 4103 | 3934 | 14.72 (.522) | .60 |
| **UK** |  |  |  |  |  |  |
| 1 | -4690.03 | 9396 | 9440 | 9415 | - - | - - |
| 2 | -4098.31 | 8230 | 8325 | 8271 | 1166.41 (< .001) | .72 |
| 3 | -3978.96 | 8009 | 8155 | 8073 | 234.84 (< .001) | .74 |
| 4 | -3963.45 | 7996 | 8193 | 8082 | 30.57 (.027) | .63 |
| 5 | -3959.39* | 8006 | 8253 | 8113 | 8.01 (1.00) | .66 |
| 6^ | - - | - - | - - | - - | - - | - - |

Note:

* models were not identified; AIC = Akaike Information Criterion; BIC = Bayesian Information Criterion; ssaBIC = sample size adjusted Bayesian Information Criterion; LMR-A = Lo-Mendell-Rubin adjusted likelihood ratio test.

index for model selection, the three-class model was selected as the best fitting model of the Irish and UK sample data.

The probabilities of accepting a COVID-19 vaccine over time in the Irish and UK samples are represented in Figs 2 and 3, respectively. In the Irish sample, class 1 included 16% of people and was characterised by extremely low probabilities of accepting a COVID-19 vaccine over time. Notably, there was a drop-off from an already low probability at Wave 1 (.15) to near zero probabilities of accepting a vaccine through Waves 2–4. This class was labelled 'Deniers'. Class 2 included 61% of the sample and was characterised by high probabilities of accepting a COVID-19 vaccine over time. Yet, it is noteworthy that the probability of acceptance steadily declined from Wave 2 (.93) to Wave 4 (.82), despite remaining high. This class was labelled 'Accepters'. Finally, class 3 included 23% of the sample and was characterised by fluctuating probabilities of accepting a COVID-19 vaccine. This class had a low-to-moderate probability of vaccine acceptance at Wave 1 (.34) that declined markedly by Wave 3 (.05) before increasing again at Wave 4 (.26). This class was labelled 'Movable Middle'.

In the UK sample, class 1 included 12% of people and was characterised by declining probabilities of accepting a COVID-19 vaccine over time. This class had a low-to-moderate probability of vaccine acceptance at Wave 1 (.32) that declined through Wave 2 (.17) and Wave 3 (.09) and remained low at Wave 4 (.10), even after the introduction of an approved vaccine. This class was labelled 'Deniers'. Class 2 included 68% of the sample and was characterised by consistently high probabilities of vaccine acceptance. Notably, the probability of vaccine acceptance rose steadily from Wave 1 (.86) to Wave 3 (.97) before decreasing at Wave 4 (.88). This class was labelled 'Accepters'. Finally, class 3 included 20% of the sample and, like class 1, demonstrated declining probabilities of vaccine acceptance from Wave 1 (.33) to Wave 2 (.12) but then diverged from class 1 as the probability of vaccine acceptance increased steadily through Wave 3 (.19) and Wave 4 (.24). This class was labelled 'Movable Middle'.

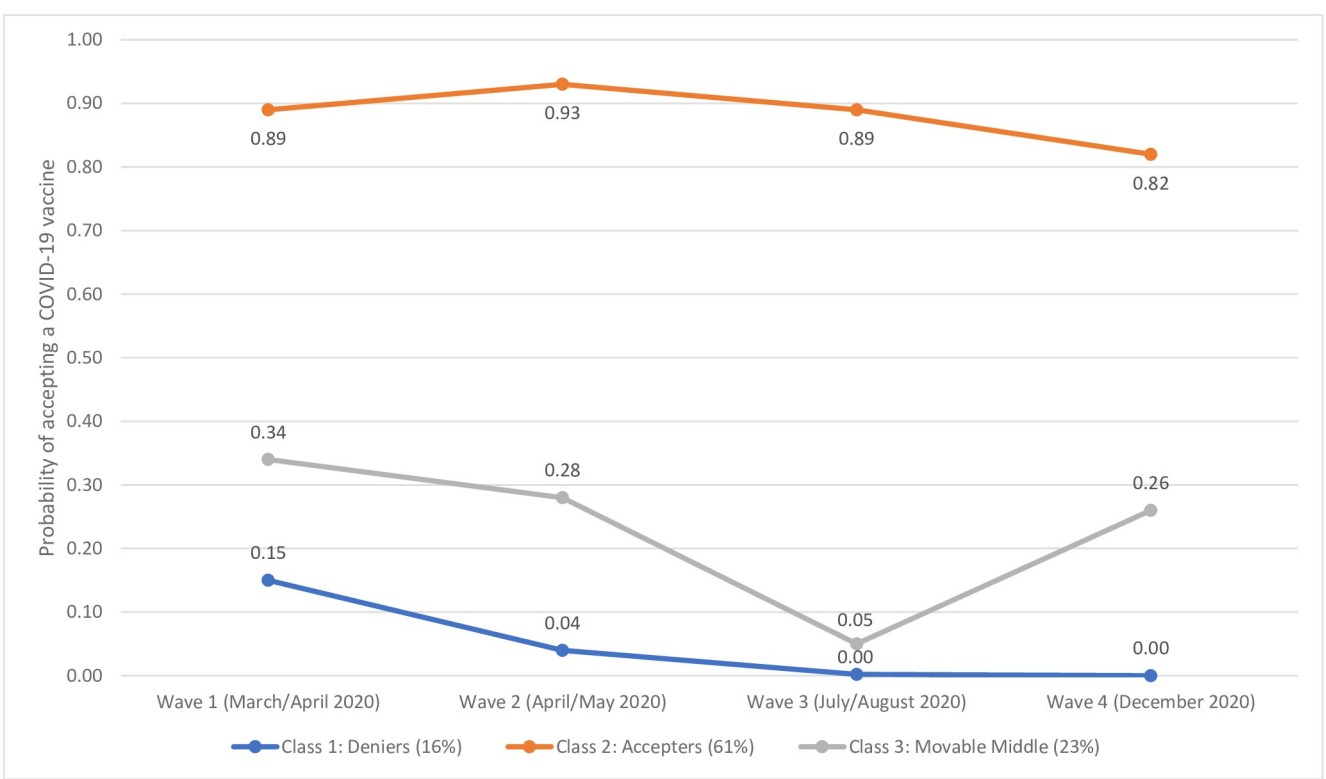

**Fig 2. Latent class probabilities of COVID-19 vaccine acceptance in the Irish sample.** Data are presented as the latent class probabilities of COVID-19 acceptance in the Irish sample (n = 1030) across four waves of data collection.

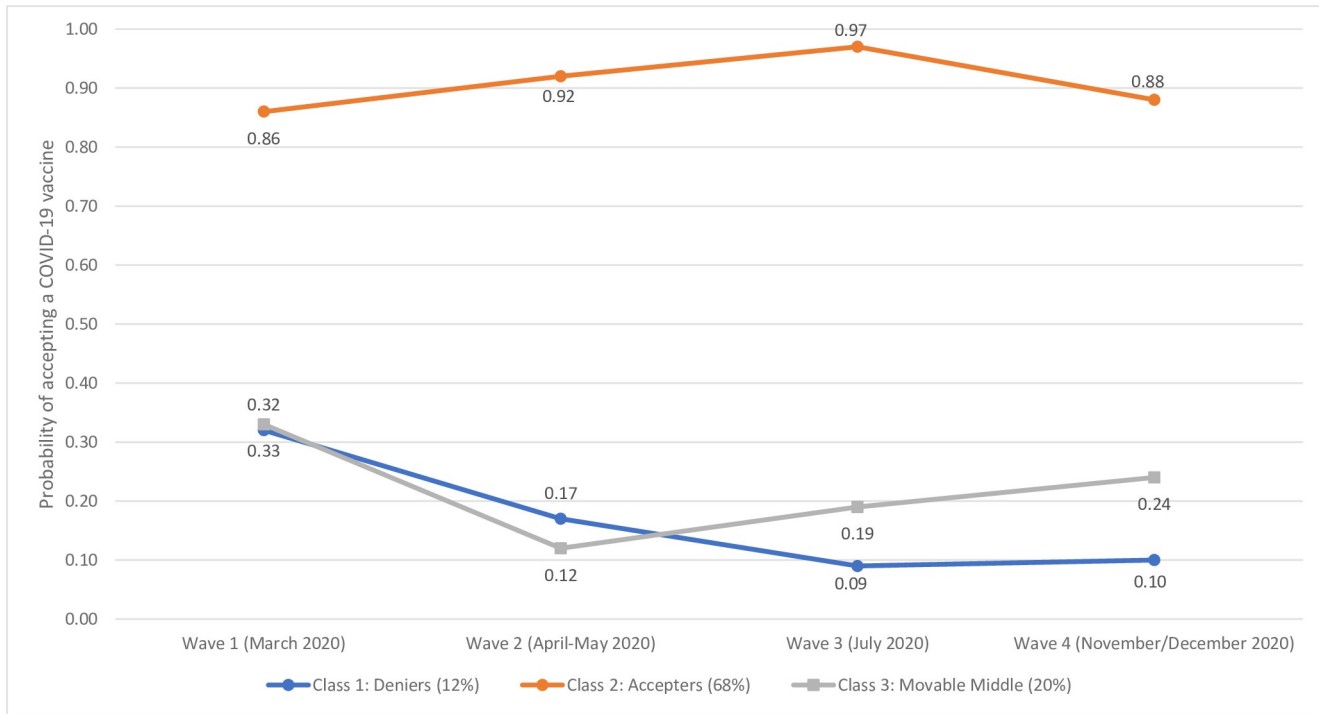

**Fig 3. Latent class probabilities of COVID-19 vaccine acceptance in the UK sample.** Data are presented as the latent class probabilities of COVID-19 acceptance in the UK sample (N = 2000) across four waves of data collection.

## Objective 3: Correlates of class membership

Based on our desire to understand why individuals were *not consistent* in their willingness to accept a COVID-19 vaccine, the class of 'Accepters' in the Irish and UK samples were set as the reference categories for analyses to determine the correlates of membership in the 'Deniers' and 'Movable Middle' classes. These findings for the Irish and UK samples are presented in Tables 4 and 5, respectively.

In the Irish sample, membership of the 'Deniers' class was significantly associated with not living with any other adults (OR = 0.40), earning less than €20,000 per year (OR = 3.96), higher levels of conspiracy mindedness (OR = 1.04), lower levels of trust in scientists and doctors (OR = 0.57), and stronger negative attitudes towards migrants in Irish society (OR = 0.87).

Membership of the 'Movable Middle' class was significantly associated with being female (OR = 2.15), lower levels of locus of control regarding the role of powerful others (OR = 0.90), higher levels of conspiracy mindedness (OR = 1.04), and lower levels of trust in scientists and doctors (OR = 0.77).

In the UK sample, membership of the 'Deniers' class was significantly associated with younger respondents (aged 18–24: OR = 16.29; 25–34: OR = 19.57; 35–44: OR = 16.16), not living with another adult (OR = 0.53), living with children under the age of 18 (OR = 1.69), abstaining from voting in the previous UK general election (OR = 2.01), lower levels of trait agreeableness (OR = 0.82), lower levels of trait neuroticism (OR = 0.82), higher levels of conspiracy mindedness (OR = 1.04), and lower levels of trust in scientists and doctors (OR = 0.72).

Membership of the 'Movable Middle' class was significantly associated with being female (OR = 1.64), being younger than 65 (aged 18–24: OR = 3.90; 25–34: OR = 3.84; 35–44: OR = 4.35; 45–54: OR = 3.08; 55–64, OR = 3.69), being of Chinese or Asian ethnicity (OR = 3.33), low weekly incomes (earning less than £300 per week: OR = 2.28; earning between £301 and £490 per week: OR = 2.08), having voted for an 'Other' party in the previous UK general election (OR = 3.25), lower levels of trait extraversion (OR = 0.90), higher levels of trait openness (OR = 1.16), and lower levels of trust in scientists and doctors (OR = 0.78).

## Discussion

Three important findings emerged from the analyses. First, the arrival of vaccines against COVID-19 coincided with a significant change in vaccine receptibility, but in only one of the two countries sampled. Second, within both samples, vaccine receptibility over time was most parsimoniously represented by three distinct groups. In Ireland and the UK, the majority of respondents belonged to a group characterised by stable acceptance that accounted for 61% and 68% of each sample, respectively. Conversely, the fewest respondents in both samples belonged to a group characterised by stable non-acceptance (Ireland: 16%) or decreasing acceptance (UK: 12%). A final group characterised by fluctuating probabilities of accepting a COVID-19 vaccine over time was also identified within each sample (Ireland: 23%; UK: 20%). Third, compared to those characterised by stable acceptance over time, individuals characterised by changing or decreasing acceptance of a COVID-19 vaccine were distinguishable, and also comparable, in relation to several individual, socio-economic, and psychological variables. The significance of these findings is described in turn below.

Compared to data that had been collected at a time when vaccine receptibility could only be considered in relation to a hypothetical vaccine (i.e., July/August 2020), data from a period when approved vaccines for COVID-19 had been introduced in both countries (December 2020) showed a significant increase in vaccine resistance in the UK, and a significant decrease in vaccine acceptance. No change in vaccine acceptance, resistance, or hesitance was identified in the Irish sample between these timepoints. The proportion of UK respondents in

**Table 4. Correlates of class membership in the Irish sample (N = 1,030).**

| | Deniers | | | Movable Middle | | |
|---|---|---|---|---|---|---|
| | **B** | ***p*** | **AOR** | **B** | ***p*** | **AOR** |
| Females | 0.34 | .341 | 1.41 | **0.76** | **.032** | **2.15** |
| 18–24 years | 1.45 | .156 | 4.26 | 0.38 | .667 | 1.47 |
| 25–34 years | 1.49 | .101 | 4.45 | 0.56 | .393 | 1.74 |
| 35–44 years | 1.35 | .119 | 3.85 | 1.08 | .069 | 2.96 |
| 45–54 years | 1.27 | .158 | 3.56 | 1.01 | .093 | 2.73 |
| [a] 55–64 years | 0.25 | .796 | 1.28 | 0.61 | .263 | 1.84 |
| Not born in Ireland | -0.16 | .794 | 0.85 | 0.27 | .615 | 1.31 |
| Non-Irish ethnicity | 1.10 | .077 | 3.01 | -0.68 | .292 | 0.51 |
| City dwelling | 0.38 | .297 | 1.46 | 0.15 | .686 | 1.16 |
| Post-secondary education | -0.24 | .560 | 0.79 | -0.09 | .818 | 0.91 |
| Unemployed | 0.42 | .299 | 1.53 | -0.08 | .856 | 0.93 |
| Religious identification | -0.29 | .475 | 0.75 | 0.35 | .449 | 1.42 |
| Living with another adult | **-0.91** | **.046** | **0.40** | -0.15 | .727 | 0.86 |
| Living with children | 0.50 | .209 | 1.65 | -0.37 | .306 | 0.69 |
| Less than €20,000 per year income | **1.38** | **.026** | **3.96** | 0.51 | .359 | 1.66 |
| €20,000–€29,999 per year income | 0.89 | .138 | 2.43 | 0.56 | .285 | 1.75 |
| €30,000–€39,999 per year income | 1.01 | .075 | 2.76 | 0.40 | .412 | 1.49 |
| [b] €40,000–€49,999 per year income | 0.92 | .183 | 2.51 | -0.16 | .792 | 0.85 |
| Physical health problem | -0.46 | .391 | 0.63 | -0.49 | .223 | 0.61 |
| Pregnant | 0.49 | .592 | 1.63 | -0.20 | .810 | 0.82 |
| COVID-19 infection—self* | - - | - - | - - | - - | - - | - - |
| COVID-19 infection—other | 0.13 | .851 | 1.14 | 0.31 | .546 | 1.37 |
| Mental health treatment | 0.08 | .839 | 1.08 | -0.75 | .057 | 0.48 |
| Chose not to vote in GE | 0.34 | .497 | 1.40 | 0.05 | .913 | 1.05 |
| Voted Sinn Fein in GE | 0.19 | .711 | 1.21 | 0.01 | .989 | 1.01 |
| Voted Independent in GE | 0.72 | .270 | 2.05 | 0.44 | .493 | 1.56 |
| [c] Voted 'Other' in GE | 0.17 | .771 | 1.19 | -0.28 | .566 | 0.76 |
| Openness | 0.18 | .146 | 1.19 | 0.03 | .728 | 1.03 |
| Conscientiousness | 0.16 | .210 | 1.18 | -0.03 | .734 | 0.97 |
| Extraversion | 0.12 | .230 | 1.12 | 0.15 | .078 | 1.17 |
| Agreeableness | -0.15 | .221 | 0.86 | -0.17 | .216 | 0.84 |
| Neuroticism | -0.15 | .172 | 0.86 | -0.04 | .674 | 0.96 |
| Locus of control—chance | -0.10 | .169 | 0.91 | 0.05 | .350 | 1.05 |
| Locus of control—powerful others | -0.05 | .363 | 0.95 | **-0.11** | **.033** | **0.90** |
| Locus of control—internal | 0.01 | .875 | 1.01 | 0.08 | .105 | 1.08 |
| Empathy | -0.02 | .408 | 0.98 | -0.01 | .755 | 0.99 |
| Conspiracy mindedness | **0.04** | **.050** | **1.04** | **0.04** | **.038** | **1.04** |
| Paranoia | 0.03 | .537 | 1.03 | 0.02 | .665 | 1.02 |
| Cognitive reflection | -0.02 | .899 | 0.98 | -0.04 | .780 | 0.96 |
| Trust in Irish state institutions | 0.01 | .856 | 1.01 | -0.01 | .833 | 0.99 |
| Trust in scientists and doctors | **-0.57** | **<.001** | **0.57** | **-0.26** | **.013** | **0.77** |
| Authoritarianism | -0.03 | .611 | 0.97 | 0.06 | .166 | 1.06 |
| Social dominance | 0.03 | .497 | 1.03 | -0.00 | .975 | 1.00 |

(*Continued*)

**Table 4.** (Continued)

| | Deniers | | | Movable Middle | | |
|---|---|---|---|---|---|---|
| | B | p | AOR | B | p | AOR |
| Attitudes toward migrants | **-0.14** | **.004** | **0.87** | -0.04 | .360 | 0.96 |

Multinomial logistic regression analyses were performed to identify the key variables associated with belonging to the 'Deniers' and 'Movable Middle' classes. All predictors are adjusted for all other covariates in the model. Note: B = unstandardized beta value; p = statistical significance value; AOR = adjusted odds ratio;

[a] = reference category is 65 year and older;

[b] reference category is €50,000 or more income;

[c] = reference category is voted for the incumbent government parties of Fine Gael or Fianna Fail;

[*] variable was not included in the model due to insufficient cases in each class; statistically significant associations (p <.05) are highlighted in bold.

November/December 2020 who indicated that they would be receptive to one of the approved vaccines for COVID-19 when it became available to them (65.5%) was slightly lower than the proportion of the sample who indicated acceptance of a hypothetical vaccine in July of the same year (71.1%). Moreover, the proportion of the sample in November/December 2020 who indicated that they would be resistant to accepting one of the approved vaccines when made available to them (15.6%) was markedly higher than the proportion who indicated resistance to a hypothetical vaccine in July 2020 (10.6%). While this trend may have been attributable to factors other than the arrival of approved vaccines (i.e., our analyses clearly indicated that fluctuation in vaccine receptibility has been at play in both countries for some time), recency of vaccine development and distribution has been identified as one of many factors that can influence vaccine hesitancy [13]. In relation to the COVID-19 pandemic specifically, a study of 1,941 Israeli healthcare workers and members of the general Israeli population has shown that the vast majority of responders' concerns were due to the assumed speed of vaccine development and related concerns surrounding quality controls [10]. It is notable that while the extant literature covers vaccine efficacy and safety extensively, and the rigorous quality controls that precede, dictate, and follow approvals [47, 48], members of the general population still identify speed, safety, efficacy, and quality control as key reasons for hesitation/concern about receiving a vaccine. It is imperative therefore that public health authorities do more to educate, inform, and intervene to challenge vaccine hesitancy on these grounds.

The current study revealed important vaccine receptibility subgroups and trends in both countries. Mixture modelling of our longitudinal data afforded a valuable opportunity to investigate (i) the proportion of each population that displayed a sustained high probability of vaccine acceptance throughout the pandemic, (ii) the proportion that displayed a sustained low probability of vaccine acceptance, and importantly, (iii) whether a 'moveable middle' group—or groups—existed, and what their receptibility profiles looked like. Overall, 61% and 68% of the Irish and UK samples, respectively, exhibited stable vaccine acceptance with acceptance probabilities in both samples above 80% across all four timepoints. However, somewhat concerningly, the trajectories for both groups ended in a downward trend. It will be important, therefore, to monitor these stable acceptance groups at later survey waves to determine what effect, if any, national vaccination programmes and communication strategies are having on acceptance levels for those who seem committed to vaccination. Notably, however, the size of these groups also reveals significant differences between countries regarding rates of acceptance and highlights the importance of country-specific approaches to understanding and tackling vaccine hesitancy and promoting vaccine receptibility.

While we expected to identify distinct subgroups in both populations characterised by low probabilities of vaccine acceptance over time, the profiles for these groups differed in

**Table 5. Correlates of class membership in the UK sample (N = 2,000).**

| | Deniers | | | Movable Middle | | |
|---|---|---|---|---|---|---|
| | B | p | AOR | B | p | AOR |
| Females | 0.32 | .209 | 1.38 | **0.49** | **.011** | **1.64** |
| 18–24 years | **2.79** | **.017** | **16.29** | **1.36** | **.003** | **3.90** |
| 25–34 years | **2.97** | **.008** | **19.57** | **1.35** | **.002** | **3.84** |
| 35–44 years | **2.78** | **.012** | **16.16** | **1.47** | **.000** | **4.35** |
| 45–54 years | 2.09 | .065 | 8.08 | **1.12** | **.004** | **3.08** |
| [a]55–64 years | 1.75 | .129 | 5.78 | **1.30** | **.000** | **3.69** |
| Not born in the UK | -0.72 | .270 | 0.49 | 0.23 | .643 | 1.25 |
| Ethnicity—White Non-UK/Irish | 0.88 | .188 | 2.40 | 0.32 | .579 | 1.38 |
| Ethnicity—Afro-Caribbean | 0.72 | .321 | 2.05 | 0.03 | .977 | 1.03 |
| Ethnicity—Chinese/Asian | -0.31 | .838 | 0.73 | **1.20** | **.031** | **3.33** |
| [b]Ethnicity—Indian/Pakistani/Bangladeshi | 0.76 | .090 | 2.13 | 0.54 | .279 | 1.72 |
| City dwelling | 0.29 | .290 | 1.33 | 0.09 | .681 | 1.09 |
| Post-secondary education | 0.25 | .334 | 1.29 | -0.09 | .652 | 0.92 |
| Unemployed | -0.61 | .167 | 0.54 | 0.34 | .164 | 1.41 |
| Religious identification | 0.30 | .303 | 1.35 | -0.02 | .907 | 0.98 |
| Living with another adult | **-0.63** | **.041** | **0.53** | -0.24 | .306 | 0.79 |
| Living with children | **0.52** | **.028** | **1.69** | 0.06 | .791 | 1.06 |
| Less than £300 per week income | **1.11** | **.023** | **3.03** | **0.82** | **.026** | **2.28** |
| £301–£490 per week income | 0.71 | .104 | 2.03 | **0.73** | **.035** | **2.08** |
| £491–£740 per week income | 0.75 | .077 | 2.12 | 0.56 | .108 | 1.75 |
| [c]£741–£1,111 per week income | 0.22 | .579 | 1.24 | 0.32 | .312 | 1.37 |
| Physical health problem | -0.29 | .379 | 0.75 | -0.33 | .186 | 0.72 |
| Pregnant | 0.46 | .309 | 1.58 | -0.43 | .488 | 0.65 |
| COVID-19 infection—self | 1.14 | .078 | 3.11 | 0.03 | .953 | 1.03 |
| COVID-19 infection—other | -1.15 | .138 | 0.32 | -0.06 | .875 | 0.95 |
| Mental health treatment | 0.23 | .378 | 1.25 | 0.09 | .660 | 1.09 |
| Chose not to vote in GE | **0.70** | **.040** | **2.01** | -0.08 | .812 | 0.93 |
| Voted Labour in GE | 0.19 | .530 | 1.21 | -0.19 | .459 | 0.83 |
| Voted Liberal Democrats in GE | -0.07 | .914 | 0.93 | -0.14 | .705 | 0.87 |
| Voted Greens in GE | -0.54 | .497 | 0.58 | -0.06 | .884 | 0.94 |
| Voted 'Nationalist' in GE | -0.13 | .867 | 0.88 | -0.24 | .638 | 0.79 |
| Voted 'Unionist' in GE | 0.52 | .290 | 1.68 | -0.09 | .842 | 0.91 |
| [d]Voted 'Other' in GE | 0.77 | .220 | 2.16 | **1.18** | **.010** | **3.25** |
| Openness | 0.04 | .611 | 1.04 | **0.15** | **.006** | **1.16** |
| Conscientiousness | -0.11 | .151 | 0.90 | -0.08 | .128 | 0.92 |
| Extraversion | 0.01 | .847 | 1.01 | **-0.11** | **.036** | **0.90** |
| Agreeableness | **-0.20** | **.012** | **0.82** | -0.10 | .103 | 0.91 |
| Neuroticism | **-0.19** | **.029** | **0.82** | -0.06 | .289 | 0.94 |
| Locus of control—chance | 0.02 | .636 | 1.02 | 0.00 | .959 | 1.00 |
| Locus of control—powerful others | -0.02 | .763 | 0.99 | -0.01 | .869 | 1.00 |
| Locus of control—internal | -0.03 | .524 | 0.97 | -0.01 | .752 | 0.99 |
| Empathy | -0.02 | .463 | 0.98 | 0.00 | .922 | 1.00 |
| Conspiracy mindedness | **0.04** | **.028** | **1.04** | 0.01 | .349 | 1.01 |
| Paranoia | 0.00 | .983 | 1.00 | -0.02 | .373 | 0.98 |
| Cognitive reflection | -0.12 | .408 | 0.89 | -0.08 | .404 | 0.92 |
| Trust in UK state institutions | 0.00 | .909 | 1.00 | -0.01 | .811 | 0.99 |

*(Continued)*

**Table 5.** (Continued)

| | Deniers | | | Movable Middle | | |
|---|---|---|---|---|---|---|
| | **B** | ***p*** | **AOR** | **B** | ***p*** | **AOR** |
| Trust in scientists and doctors | **-0.33** | **.001** | **0.72** | **-0.26** | **<.001** | **0.78** |
| Authoritarianism | 0.05 | .265 | 1.05 | 0.01 | .682 | 1.01 |
| Social dominance | 0.04 | .150 | 1.04 | 0.01 | .440 | 1.01 |
| Attitudes toward migrants | -0.05 | .220 | 0.96 | -0.04 | .127 | 0.96 |

Multinomial logistic regression analyses were performed to identify the key variables associated with belonging to the 'Deniers' and 'Movable Middle' classes. All predictors are adjusted for all other covariates in the model. Note: B = unstandardized beta value; p = statistical significance value; AOR = adjusted odds ratio;

[a] = reference category is 65 year and older;

[b] = reference category is 'White British or Irish';

[c] = reference category is £1,112 per week or more;

[d] = reference category is voted for the incumbent Conservative government party; statistically significant associations (p < .05) are highlighted in bold.

important ways. While the Irish sample included a group characterised by sustained low-to-near zero probabilities of acceptance at each survey wave (16%), the UK's most resistant group (12%) began with a 32% probability of acceptance that steadily declined to 10% by Wave 4. The Irish non-acceptance group, therefore, reflected more extreme and stable resistance compared to those who were most resistant in the UK. Several studies have shown that upwards of approximately 10% of study populations appear to be opposed to vaccinations in whatever form they take [49, 50]; therefore, these findings were not entirely surprising. It was notable, however, that resistance was lowest in both countries at the beginning of the pandemic (~6–10% in March/April 2020), and that this resistance steadily rose (significantly between some survey waves) to ~16–18% by Waves 3 (July/August 2020) and 4 (November/December 2020). Resistance to actual approved vaccines in December 2020, therefore, was concerningly high. If resistance remains at this level or continues to rise, public health officials will likely need to consider how to reach and persuade a now substantial subpopulation that has traditionally been shown to be extremely resistant to vaccine promotional campaigns and public health messaging regarding inoculation generally [51, 52].

A third group was also identified in both countries. This group was considered to represent a 'moveable middle' or 'changing' group that may hold important significance for future public health initiatives that seek to achieve herd-protection against SARS-CoV-2. In the Irish sample, this group was characterised by a 26% probability of accepting a vaccine in December 2020 when approved vaccines had been developed. However, in the months preceding vaccine development (July/August 2020), this same group of respondents exhibited only a 5% probability of acceptance, while at the beginning of the pandemic, acceptance probability was at its highest (34%). Comparatively, the 'moveable middle' group in the UK sample exhibited a similar probability of acceptance in November/December 2020 (24%), and at the beginning of the pandemic (33%) but had its lowest level of acceptance in April/May 2020 (12%). These groups have fluctuated in their positions over the duration of the pandemic, and while there may be cause for optimism in the upward trends identified at the most recent data collection time-points, it must be noted that neither of these groups displayed a probability of acceptance above 34% at any time since the beginning of the pandemic.

While the extant research literature details many distinct socio-demographic and psychological indicators of vaccine hesitancy generally [6, 53, 54], and a burgeoning literature has begun to list those common to COVID-19 vaccines specifically [10, 16, 55], studies describing characteristics associated with stability or change in vaccine receptibility over time are lacking.

Our findings revealed important similarities and distinctions in vaccine receptibility between those in the 'movable middle' and those characterised by stable resistance in both countries.

First, those who fluctuated in their receptiveness to a COVID-19 vaccine in Ireland and the UK were more likely to be female and to lack trust in scientists and health care professionals. Evidence suggests that, in relation to COVID-19 vaccination specifically, females may have concerns surrounding issues such as fertility and pregnancy [56, 57]. As has been highlighted earlier, trust in scientists and health care professionals (particularly regarding the speed of vaccine development and distribution) seems also to be of particular concern for many who are hesitant about a COVID-19 vaccine specifically [10]. Public health messaging, therefore, tailored specifically to allay concerns and/or fears that may be specific to women, and/or to educate and reassure the public about quality controls and standards relating to the development, distribution, administration, and review of COVID-19 vaccines may prove useful. Notable distinctions were also evident for the moveable middle groups across samples. In Ireland, those who fluctuated over time were more likely than accepters to believe that powerful others were responsible for their experiences and to hold conspiratorial beliefs, while those in the UK were more likely than accepters to be younger, of Chinese/Asian ethnicity, have a lower level of income, have voted 'other' in the last general election, be lower in extraversion, and higher in openness. These distinct country specific characteristics may help to further inform and refine public health messaging in ways that are contextually sensitive to each population.

Second, those who remained resistant over time in Ireland and the UK tended not to live with any other adults, to hold conspiratorial beliefs, and to lack trust in scientists and health care professionals. While those who remained resistant over time may be more challenging to reach or persuade than those who fluctuate in their receptibility, these common indicators of resistance may prove useful in informing our understanding of who these people are and why they are susceptible and committed to the beliefs they hold. Individuals living alone have been shown to lack important opportunities to explore/discuss their concerns or to reality test their assumptions about the world in which they live [58, 59], while those who are open/receptive to conspiratorial interpretations of world events often dismiss information sourced from or disseminated by traditional, scientific and/or authoritative sources [60, 61]. Notably, as was also evident for the change groups, stable resisters in both countries also differed in specific ways. In Ireland, these individuals were uniquely characterised by low income and negative views towards migrants, while in the UK, those most resistant to a COVID-19 vaccine were more likely to have children, not to have voted in the last general election, and to be lower in the personality traits of agreeableness and neuroticism. Each of these indicators has previously been shown to be associated with vaccine hesitancy/resistance [6, 62]. That they do not predict resistance in the same way within different populations and in relation to common vaccines likely reflects the context specific complexity of vaccine hesitancy as a phenomenon and the challenging terrain that must be navigated by those seeking to tackle it.

These findings should be interpreted considering several limitations. First, non-probability quota-based sampling methods were used to recruit samples via the Internet. This opt-in mode of recruitment employed by the survey company who facilitated the data collection (Qualtrics), albeit being a cost-effective method for gaining fast access to a large and diverse sample (and the most feasible method of recruitment during the pandemic), inevitably meant that it was not possible to know if participants in these panels differed in important ways from members of the public that do not belong to the panels. Second, the current study was also limited to two western, European countries whose populations had many social, cultural, economic, and political similarities. However, while these populations may have been similar in many respects, our findings highlight notable differences between countries in relation to (i) the proportions of each population that were receptive, hesitant, and resistant over time, (ii)

the profiles and trajectories of these groups, and (iii) the specific indicators that predicted fluctuation and stable resistance over time. Now that vaccination programmes are underway in many countries, our findings highlight the importance of population-specific analyses of vaccine hesitancy and the continued monitoring of this phenomenon as vaccination programmes advance. Relatedly, the extent to which these results will generalise to other nations is unknown. It is essential that other (low, middle, and high income) countries obtain estimates of change in hesitancy/resistance to COVID-19 vaccination in their general populations, given that vaccination efforts will only succeed if sufficiently undertaken globally. Third, while the use of nationally representative samples from two countries is a key strength, these samples are representative of general adult populations and do not include members of the public that are institutionalised (e.g., hospital care, prisons, refugee centres) or difficult to reach (e.g., those not online, the homeless, etc.). The inability to survey these members of society also limits the generalisability of our results.

## Conclusion

Our findings suggest that approximately two-thirds of adults in the general populations of the UK and Ireland had consistently high probabilities of accepting a COVID-19 vaccine during the first nine months of the global pandemic. To achieve wider vaccine coverage, it will be important to reach the 20–25% of people in society who belong to the so-called 'moveable middle'. In both samples, these individuals were more likely to be women, and to have lower levels of trust in scientists, doctors, and other healthcare professionals. Furthermore, context-specific identifiers were also evident such as younger age, Asian ethnicity, and lower income in the UK, and conspiracy mindedness and external locus of control in Ireland. These findings can be used to aid public health efforts in both countries to reach those in society whose minds can be changed with regards to COVID-19 vaccination.

## Author Contributions

**Conceptualization:** Philip Hyland, Frédérique Vallières, Jamie Murphy.

**Formal analysis:** Philip Hyland, Frédérique Vallières, Jamie Murphy.

**Writing – original draft:** Philip Hyland, Frédérique Vallières, Todd K. Hartman, Ryan McKay, Sarah Butter, Jamie Murphy.

**Writing – review & editing:** Philip Hyland, Frédérique Vallières, Todd K. Hartman, Ryan McKay, Sarah Butter, Richard P. Bentall, Orla McBride, Mark Shevlin, Kate Bennett, Liam Mason, Jilly Gibson-Miller, Liat Levita, Anton P. Martinez, Thomas V. A. Stocks, Thanos Karatzias, Jamie Murphy.

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
