## [Decision Letter · Decision Letter 0]

13 Aug 2021

PONE-D-21-23711

Detecting and describing stability and change in COVID-19 vaccine receptibility in the United Kingdom and Ireland

PLOS ONE

Dear Dr. Butter,

Thank you for submitting your manuscript to PLOS ONE. After careful consideration, we feel that it has merit but does not fully meet PLOS ONE’s publication criteria as it currently stands. Therefore, we invite you to submit a revised version of the manuscript that addresses the points raised during the review process.

We look forward to receiving your revised manuscript.

Kind regards,

Jun Tanimoto

Academic Editor

PLOS ONE

Journal Requirements:

"I have read the journal's policy and the authors of this manuscript have the following competing interests: Richard Bentall reports financial support was provided by UK Research and Innovation. Philip Hyland reports financial support was provided by Health Research Board."

We note that you received funding from a commercial source: "UK Research and Innovation and Health Research Board"

Reviewers' comments:

Reviewer's Responses to Questions

**Comments to the Author**

1. Is the manuscript technically sound, and do the data support the conclusions?

Reviewer #1: Yes

Reviewer #2: Yes

2. Has the statistical analysis been performed appropriately and rigorously? 

Reviewer #1: Yes

Reviewer #2: Yes

3. Have the authors made all data underlying the findings in their manuscript fully available?

Reviewer #1: Yes

Reviewer #2: Yes

4. Is the manuscript presented in an intelligible fashion and written in standard English?

Reviewer #1: Yes

Reviewer #2: Yes

5. Review Comments to the Author

Reviewer #1: Because of the hotness of the focal topic; about the acceptance of a COVID-19 vaccination, the report work might be informative and meaningful to the world-wide audience. The science of their survey in terms of ethics, procedure, statistics, and other analysis, looks healthy. Therefor, I evaluate the work with a positive impression.

The suggestions are given as below to improve the final form of contribution.

#1.

The main result about the acceptance of each of those regions (two nations) and its time-series in the real wake of COVID-19 is quite interesting. If it is possible, the audience including me would love to hear why the acceptance level amid those two countries being different, which can be referring to unvisitable, potential and social reasons in view of comparison between those two close neighboring countries (closes each other) .

#2.

The acceptance of vaccination, which might be quantified by each of responders with reference to various factors, as the authors discussing, such as benefic (vis-à-vis thread of infection), risk of side-effect, and other things, should be more profoundly discussed. One important aspect is whether such acceptance of vaccination can be predicted by a model framework in the help from applied mathematical framework such as Evolutionary Game Theory and others. Citing several precursors; for example, ‘Hypothetical assessment of efficiency, willingness-to-accept and willingness-to-pay for dengue vaccine and treatment: a contingent valuation survey in Bangladesh, Journal Human Vaccines & Immunotherapeutics, 2020’, the authors should supplement further discussion.

Reviewer #2: This paper is well written, the topic is interesting, and the results seem correct; the work is acceptable. I make some recommendations for minor revisions. Besides these issues, the work appears methodologically sound and is well written.

##1

Your abstract looks very technical level, and it isn't easy to understand the key point of your findings. I want to ask you to rewrite the abstract. For instance, the abstract describes the framework but leaves it to the reader to dig into the Results to find out what is novel about the predictions. I think the work would be better served if the abstract highlighted some of the most important findings emerging from this model. Or, what novel/surprising results emerged from this work?

##2

If possible, please add one figure that can represent overall picture of study settings and findings for general readers.

##3

The current form of figure is terrible, less-explanation, unclear legend, unsmooth colored and the range of parameters. Would you please make clear the figures?

Figure captions: please re-write the figure caption in detail that can easily be persuasive to a reader.

##4

In the introduction, authors should cite the following literatures,

Hypothetical assessment of efficiency, willingness-to-accept and willingness-to-pay for dengue vaccine and treatment: a contingent valuation survey in Bangladesh, Human vaccine and Immunotherapeutics, DOI: 10.1080/21645515.2020.1796424 (2020).

Evolutionary game theory modelling to represent the behavioural dynamics of economic shutdowns and shield immunity in the COVID-19 pandemic. R. Soc. Open Sci. 7: 201095. http://dx.doi.org/10.1098/rsos.201095 (2020).

A cyclic epidemic vaccination model: Embedding the attitude of individuals toward vaccination into SVIS dynamics through social interactions, Physica A, 581, 126230 (2021).

6. PLOS authors have the option to publish the peer review history of their article (what does this mean?). If published, this will include your full peer review and any attached files.

Reviewer #1: No

Reviewer #2: No

---

## [Author Response · Author response to Decision Letter 0]

22 Aug 2021

See attached 'Response to Reviewers' document.

---

## [Decision Letter · Decision Letter 1]

7 Oct 2021

Detecting and describing stability and change in COVID-19 vaccine receptibility in the United Kingdom and Ireland

PONE-D-21-23711R1

Dear Dr. Butter,

We’re pleased to inform you that your manuscript has been judged scientifically suitable for publication and will be formally accepted for publication once it meets all outstanding technical requirements.

Kind regards,

Jun Tanimoto

Academic Editor

PLOS ONE

Additional Editor Comments (optional):

Reviewers' comments:

Reviewer's Responses to Questions

**Comments to the Author**

1. If the authors have adequately addressed your comments raised in a previous round of review and you feel that this manuscript is now acceptable for publication, you may indicate that here to bypass the “Comments to the Author” section, enter your conflict of interest statement in the “Confidential to Editor” section, and submit your "Accept" recommendation.

Reviewer #1: All comments have been addressed

Reviewer #2: (No Response)

2. Is the manuscript technically sound, and do the data support the conclusions?

Reviewer #1: Yes

Reviewer #2: (No Response)

3. Has the statistical analysis been performed appropriately and rigorously? 

Reviewer #1: Yes

Reviewer #2: (No Response)

4. Have the authors made all data underlying the findings in their manuscript fully available?

Reviewer #1: Yes

Reviewer #2: (No Response)

5. Is the manuscript presented in an intelligible fashion and written in standard English?

Reviewer #1: Yes

Reviewer #2: (No Response)

6. Review Comments to the Author

Reviewer #1: The authors seems almost responding the suggestions I gave.

Honest, I found that some of the points were ignored, which is bit pitty.

Reviewer #2: (No Response)

7. PLOS authors have the option to publish the peer review history of their article (what does this mean?). If published, this will include your full peer review and any attached files.

Reviewer #1: No

Reviewer #2: No

---

## [Editor Report · Acceptance letter]

12 Oct 2021

PONE-D-21-23711R1 

Detecting and describing stability and change in COVID-19 vaccine receptibility in the United Kingdom and Ireland 

Dear Dr. Butter:

I'm pleased to inform you that your manuscript has been deemed suitable for publication in PLOS ONE. Congratulations! Your manuscript is now with our production department. 

Kind regards, 

on behalf of

Prof. Jun Tanimoto 

Academic Editor

PLOS ONE